# Comparison of Non-Contrast CT vs. Contrast-Enhanced CT with Both Intravenous and Rectal Contrast Application for Diagnosis of Acute Colonic Diverticulitis: A Multireader, Retrospective Single-Center Study

**DOI:** 10.3390/diagnostics15010029

**Published:** 2024-12-26

**Authors:** Dorottya Móré, Stella Erdmann, Arved Bischoff, Verena Wagner, Hans-Ulrich Kauczor, Lukas F. Liesenfeld, Katharina Abbasi Dezfouli, Athanasios Giannakis, Miriam Klauß, Philipp Mayer

**Affiliations:** 1Diagnostic and Interventional Radiology, Heidelberg University Hospital, 69120 Heidelberg, Germany; dorottya.more@med.uni-heidelberg.de (D.M.); arved.bischoff@med.uni-heidelberg.de (A.B.); verenamaria.wagner@med.uni-heidelberg.de (V.W.); hans-ulrich.kauczor@med.uni-heidelberg.de (H.-U.K.); katharina.abbasidezfouli@med.uni-heidelberg.de (K.A.D.); a-giannakis@hotmail.com (A.G.); miriam.klauss@med.uni-heidelberg.de (M.K.); 2Institute of Medical Biometry, University of Heidelberg, 69120 Heidelberg, Germany; erdmann@imbi.uni-heidelberg.de; 3Department of General, Visceral and Transplantation Surgery, Heidelberg University Hospital, 69120 Heidelberg, Germany; lukas.liesenfeld@med.uni-heidelberg.de

**Keywords:** colonic diverticulitis, computed tomography, non-contrast computed tomography, contrast enema, reader study

## Abstract

**Objectives**: To evaluate the non-inferiority of non-contrast CT compared to contrast-enhanced CT with both intravenous and rectal contrast application for the diagnosis of acute colonic diverticulitis. **Methods**: Five readers retrospectively evaluated the non-contrast and contrast-enhanced series of CTs of 205 consecutive patients with clinical suspicion of acute diverticulitis. Two randomized reading sessions, both containing all 205 cases as either contrast-enhanced or non-contrast (1:1) series, were performed with ≥8 weeks washout between them. The non-inferiority margin was set to 0.1. **Results**: The pooled prevalence (all readers) of diverticulitis was similar for non-contrast CT (63.9%, range: 60.5–65.0%) and contrast-enhanced CT (64.4%, 61.5–67.8%). Non-contrast CT was non-inferior for the diagnosis of diverticulitis (accuracy 0.90 [95% confidence interval: 0.89, 0.92]) compared to contrast-enhanced CT (0.92 [0.90, 0.94]; the difference in accuracy: −0.01 [−0.04, 0.01]) (normal deviate test: *p*-value_one-sided_ = 5.20 × 10^−6^). Sensitivities for perforation and abscess were slightly but significantly lower for the non-contrast CT than for the contrast-enhanced CT (differences: −0.15 [−0.20, −0.05], −0.17 [−0.27, −0.07]), while no differences in accuracies and specificities were observed. **Conclusions**: Non-contrast CT is non-inferior to contrast-enhanced CT (intravenous and rectal contrast) for the diagnosis of acute colonic diverticulitis. Contrast-enhanced CT is associated with significantly higher sensitivities for the presence of an abscess or perforation.

## 1. Introduction

Acute colonic diverticulitis is a common cause of abdominal pain in the emergency departments with increasing incidence [1].

Clinical assessment and laboratory results are considered insufficient for the diagnosis of acute diverticulitis [2]. CT has been named as the modality of choice for the accurate assessment of the disease and complications by most guidelines, including the Danish national guideline [3], the British National Institute for Health and Care Excellence (NICE) guideline [4], the European Society of Coloproctology guideline [5], and others [6,7]. Few guidelines, such as a Dutch guideline, recommend a step-up strategy with ultrasound (US) performed by an expert operator as a first-line imaging modality, as it does not involve ionizing radiation, followed by CT in case of inconclusive ultrasound results [8,9]. However, since the US tends to underestimate the disease severity [10,11], as highlighted in the German national S3 guideline [12], the US is more frequently seen as a backup modality in case of contraindications for CT [4], e.g., in pregnant or possibly pregnant women [3]. Magnetic resonance imaging is not recommended by the majority of guidelines due to insufficient data [7,12].

However, there is no universal agreement on the CT examination protocol for suspected colonic diverticulitis. Before the advent of CT, fluoroscopy with contrast enema was used to visualize the leakage of contrast at the site of perforation [13]. Analogously, a preparatory rectal contrast enema has long been considered part of the CT protocol [14,15,16,17]. The long-term practice of intravenous contrast administration is mostly based on expert opinions [18].

The higher examination time, dose, costs, potential adverse effects associated with contrast application, and improved image quality due to innovations in CT technology have prompted an investigation into the applicability of simplified and non-contrast imaging protocols.

Among diverticulitis guidelines, there is disagreement regarding the use of any contrast agent application for CT diagnosis of suspected diverticulitis [19]. Older guidelines especially advocate the use of CT with both intravenous and rectal contrast applications [3,6,20,21,22]. More recent guidelines discuss the use of contrast agents more critically: The revised American College of Radiology Appropriateness Criteria from 2023 point out that although CT with intravenous contrast is commonly used for the diagnosis of acute diverticulitis due to a potentially improved characterization of the bowel wall, the application of contrast agent might not be necessary for most patients [7]. The most recent German S3 guideline considers the level of evidence of these studies not yet sufficient for the routine omission of intravenous and rectal contrast for diagnosing acute diverticulitis in the absence of contraindications [12].

This study aimed to evaluate whether the assessment of suspected cases of sigmoid diverticulitis using non-contrast CT is of inferior diagnostic value compared to CT using both intravenous and rectal contrast.

## 2. Materials and Methods

Patients: Consecutive CT scans of adults (≥18 years old) performed between July 2017 and June 2022 for suspected sigmoid diverticulitis, including both relevant CT series, were retrospectively collected from the local radiology information system (*n* = 245). The exclusion criteria are summarized in Figure 1.

CT acquisition: All CT examinations in this study included a non-contrast CT series of the abdomen followed by a portal venous phase series obtained after administration of a mixture of contrast agent and water through a rectally introduced common urinary catheter. The CTs were performed on four different CT machines: Somatom Edge Plus, Siemens Healthineers AG (Erlangen, Germany, 64 detector rows, *n* = 109); Somatom Definition Force, Siemens Healthineers AG (2 × 192 detector rows, *n* = 60); Philips Brilliance iCT 256, Philips Healthcare (Amsterdam, The Netherlands, 256 detector rows, *n* = 8); Philips iQon 7500, Philips Healthcare (128 detector rows, *n* = 28) in craniocaudal direction. Axial reconstructions with a slice thickness of 3 mm (slice interval 1.5–2 mm) in a medium soft reconstruction algorithm were performed for the visual analysis.

Intravenous contrast: 70–100 mL (ca. 0.1 mL/kg) of the iodine-based intravenous contrast agent Accupaque (350 mg J/mL; GE HealthCare, Chicago, IL, USA), 70–90 s delay, 2–3 mL/s injection rate. Intraluminal contrast: a mixture of contrast agent Accupaque (350 mg J/mL) and water (1:20), 120–700 mL according to the patient’s tolerance.

All CTs were completed according to our in-house standard operating procedures concerning contrast administration. Intravenous hydration before or hemodialysis following CT due to a preexistent renal insufficiency was documented in 2-2 cases, respectively. Two patients received H1–H2-receptor blockers due to a positive history of contrast allergy, and two patients received a sodium perchlorate solution due to latent hyperthyroidism detected in their laboratory results.

Radiological analysis: Two board-certified radiologists (8 years of experience, ‘experts’) and three radiology residents (3, 4, and 5 years of experience, ‘non-experts’) completed separate readings of the non-contrast and the contrast-enhanced (contrast-enhanced) series of all CT studies. Two reading sessions were completed with a minimum of 8 weeks washout between them, both containing all 205 cases as either contrast-enhanced or non-contrast series (1:1). The allocation of the contrast-enhanced and non-contrast series to the first or second reading session was randomized, and the sequence of the cases per session was individual for each reader.

The CT series were presented in anonymized form, without any clinical information, in the Philips IntelliSpace Portal (Version 12.1, Philips Healthcare) on a Picture Archiving and Communication System (PACS) workstation, where automatic multiplanar reconstructions were available in addition to the original transversal series. Readers were allowed to adjust window settings according to their individual preferences.

The presence of diverticulitis, the affected bowel segment, contained or free peritoneal perforation of the bowel, and abscess (diverticulitis-associated or not) were assessed in both non-contrast and contrast-enhanced series. In the contrast series, the readers also assessed whether the intraluminal contrast reached the localization of the pathological changes in the bowel and if an extralumination of the contrast could be detected. In the absence of diverticulitis, alternative acute abdominal pathologies were documented as the suspected cause of the clinical symptoms.

Reference standard: The original reports were obtained from the local radiology information system and considered the reference standard. This could be overruled by the histological diagnosis and by the final diagnosis as stated in the electronic medical record ≤7 days after imaging. In equivocal cases and/or incomplete descriptions in the original report, a consensus read by a team of two senior radiology consultants (11–19 years of experience) was considered the reference standard (Figure 2).

Statistical analysis: Statistical data analysis was performed using the statistical programming language R (R Foundation for Statistical Computing, version 4.2.1, Vienna, Austria).

As a primary analysis, the non-inferiority of non-contrast CT to contrast-enhanced CT for the accuracy of diagnosing sigmoid diverticulitis was tested by the non-inferiority test suggested by Saeki and Tango [23] for the one-sided significance level of 0.025. As recommended by Ahn et al., the non-inferiority margin was defined before data acquisition [24]. Our non-inferiority margin of 10% was chosen smaller than the difference in diagnostic accuracy without imaging, 70.5–86% [21,25,26], and with contrast-enhanced CT imaging, up to 99% [14,27] from previous studies (99–86% = 13%; 99–70.5% = 28.5%).

All *p*-values except that of the primary analysis are to be interpreted in a descriptive manner. *p*-values < 0.05 were considered statistically significant. There are no missing values in the data. No missing values were imputed. Additional information is provided in the Appendix A.

## 3. Results

A total of 205 patients were included in the study (103 women, 102 men, mean age 60.0 ± standard deviation 13.6 years).

### 3.1. Diagnosis of Diverticulitis, Perforation, and Abscess—Radiologists’ Diagnostic Performance and Interrater Agreement

The prevalence of diverticulitis in the study population in the reference standard was 136 out of 205 cases (66%). The prevalences of diverticulitis, perforation, and abscess in the reference standard and according to the readers are summarized in Table 1. The classification of the cases according to the (modified) Hinchey classification is presented in Appendix A [28].

The values of accuracy, sensitivity, and specificity for the diagnosis of diverticulitis (single readers and all readers combined) were similar for non-contrast and contrast-enhanced CTs (Table 2; Figure 3). Values for accuracy, sensitivity, and specificity ranged from 0.878 to 0.922, 0.882 to 0.941, and 0.841 to 0.942 for non-contrast CT and from 0.902 to 0.942, 0.897 to 0.949, and 0.841 to 0.942 for contrast-enhanced CT.

Interrater agreement for the diagnosis of diverticulitis was substantial for both non-contrast and contrast-enhanced CT (Table 3).

The normal deviate test to test non-inferiority for non-contrast to contrast-enhanced CT by means of the accuracy detecting colonic diverticulitis has a score Z value of 4.408695 [23] (one-sided *p*-value = 5.20 × 10^−6^). As 5.20 × 10^−6^ < 0.025, non-inferiority of non-contrast versus contrast-enhanced CT can be claimed for a non-inferiority margin of 0.1 in the primary analysis.

For the diagnosis of perforation (all perforations pooled) and abscess, the pooled reader diagnostic sensitivity was lower in non-contrast than in contrast-enhanced CT, but specificity and accuracy were similar. A separate analysis of contained and free perforation showed a lower sensitivity of non-contrast CT for contained perforation but not for free peritoneal perforation (95% confidence interval included the 0; Table 2). Interrater agreement for perforation was moderate for both non-contrast and contrast-enhanced CT, while for abscess, it was fair for non-contrast and moderate for contrast-enhanced CT (Table 3). The size and ratings of abscesses are summarized in Appendix A.

Expert readers had a higher specificity for perforation (all perforations pooled and separate analysis for contained and free perforation) and abscess and a higher accuracy for free perforation than non-expert readers (Appendix A).

Estimations of additional time expenditures for contrast-enhanced CT scans compared to non-contrast CT scans, based on our practical experience and empirical values from the literature [30,31,32,33], are given in the Appendix A.

Examples of diverticulitis cases are presented in Figure 4.

### 3.2. Technical Assessment of Contrast Enema

The reference standard and readers alike detected sufficient intraluminal contrast at the site of a bowel pathology in less than a quarter of cases (24%; 22–24%, respectively). Their assessment of contrast extralumination was strongly discordant (16%; 2–29%) (Appendix A).

### 3.3. Alternative Diagnoses

For expert readers, the rate of correct identification of alternative diagnoses was not statistically different between the contrast-enhanced CT and the non-contrast CT read (*p* = 0.56), but for non-expert readers, contrast application was associated with a significantly improved rate of correctly recognized alternative diagnoses (*p* = 0.01) (Table 4; Appendix A). Example cases with alternative diagnoses are presented in Appendix A.

## 4. Discussion

The present single-center, multireader CT study focuses on the diagnostic necessity of intravenous and intraluminal contrast for the diagnosis of colonic diverticulitis. The availability of both a non-contrast CT series and a series with intravenous as well as intraluminal contrast (only minutes apart) for every patient allows for a powerful intra-patient comparison. This study shows that the diagnostic value of non-contrast CT for the diagnosis of acute diverticulitis is overall not inferior to CT series with intravenous and intraluminal contrast.

Some authors and guidelines still support the use of positive rectal contrast in CT for suspected colonic diverticulitis [10,12,21,34]. DeStigter et al. argued that rectal contrast could improve colonic distension and prevent a false appearance of wall thickening [34], hampering the diagnosis of diverticulitis [15]. In a recent dual-reader study with inter-patient comparison of CT scans with or without rectal contrast and each +/− intravenous contrast, CT without positive rectal contrast was non-inferior compared to CT with rectal contrast for diagnosis of acute diverticulitis [35]. In another inter-patient comparison study by Meyer et al., where a minority of patients had received rectal contrast, it did not improve the accuracy of CT assessment [17]. The potentially more powerful and more accurate intra-patient comparison used in our study is made possible by the availability of both scans with and without rectal contrast in each patient. Our results confirm that CT diagnosis of diverticulitis using non-contrast CT is non-inferior to CT with both rectal and intravenous contrast. These findings from our study and the study by Meyer et al. suggest that the appraisal of morphological changes detectable without rectal contrast filling is sufficient for the diagnosis of diverticulitis [15].

Another argument in favor of positive rectal contrast is that extralumination of contrast agents may facilitate the diagnosis and localization of colonic perforation in patients with diverticulitis [6,36]. In the era of fluoroscopic contrast enema, extraluminal contrast agent was seen as the most reliable sign of diverticulitis by some authors [18]. In the present study, sufficient intraluminal contrast was rarely achieved at the site of the perforation, possibly due to an insufficient anal sphincter tone or patient discomfort [15,37]. Notably, even with adequate intraluminal contrast, the rate of extralumination at the site of perforation was low and reported inconsistently by the readers. Yet, the correct diagnosis was usually warranted by other signs of perforation, such as extraluminal air and/or stool. This is in agreement with previous studies and a meta-analysis with low rates of contrast extralumination in colonic injuries/perforations [37], perforated diverticulitis [15], and alimentary tract perforation [38]. These and our findings suggest a limited relevance of endoluminal contrast application and extralumination in the diagnostics of perforation.

The use of intravenous contrast in CT for suspected diverticulitis is still regarded as the standard of care by many authors and guidelines [3,10,12,21,22,36,39].

Synder argues that intravenous contrast can improve the differentiation of diverticulitis from colorectal cancer [18], which is prevalent in 1.9% of patients with suspected acute diverticulitis [40]. However, differentiation between CRC and diverticulitis is of limited accuracy even with contrast-enhanced CT [41]. In the present study, the two cases of sigmoid carcinoma diagnosed during subsequent colonoscopy were already indicated in the initial radiology report and were reported by the majority of readers in both contrast-enhanced and non-contrast CT.

Another concern is that the omission of intravenous contrast may compromise the detection of extracolonic diverticulitis manifestations, especially pericolonic abscesses, in the absence of ring enhancement [42]. In line with previous studies, we observed that the accuracy of the diagnosis of abscesses in non-contrast CT is similar to contrast-enhanced CT [43], while sensitivity was lower in the non-contrast CT [44]. Although an abscess has potential implications for treatment planning (antibiotics, CT-guided drainage), the undetected abscesses in the present and previous studies were rather small (median size 3–4 cm) [44] (Appendix A). In accordance with the relevant guidelines, which consider abscesses >3 cm [3,5] or 4–5 cm [6,19,22] as relative indications for invasive treatment, all abscesses missed on non-contrast CT in our study were treated conservatively. Notably, both non-contrast and contrast-enhanced CT were highly sensitive for detecting free peritoneal perforations that require immediate surgery [45].

Previous studies have shown partly conflicting results regarding the advantages of CT with intravenous contrast for diagnosing abdominal pathologies other than acute diverticulitis. Some studies found non-contrast CT alone is accurate for the diagnosis of acute abdominal conditions [46]. Shaish et al., however, recently reported the accuracy of unenhanced CT to be approximately 30% lower than that of contrast-enhanced CT for the evaluation of acute abdominal pain in the emergency department [47]. In previous CT diverticulitis studies, CT without intravenous contrast allowed for the correct diagnosis of alternative conditions other than diverticulitis in 88% (low-dose CT) [44] and 78% of cases (CT with colonic contrast only) [16]. Similarly, in our study, most alternative diagnoses were identified on non-contrast CT in at least two-thirds of cases. However, acute pathologies of the mesenteric vessels often remained undetected in the absence of intravenous contrast (Appendix A and Appendix A). Therefore, the use of intravenous contrast is imperative when there is clinical suspicion of mesenteric ischemia to avoid delay in diagnosis [48,49]. Moreover, if septic thrombosis of the portal vein (pylephlebitis) is suspected in patients with abdominal infections (diverticulitis or non-diverticulitis), i.v. contrast can be beneficial [50].

While the current study design precludes a separate evaluation of the potential advantages of intravenous contrast and rectal contrast, we assume that the slight diagnostic superiority of the contrast-enhanced CT read over the non-contrast CT read is attributable to the intravenous and not to the rectal contrast application. Intravenous contrast can aid the classification of a fluid collection as an abscess due to rim enhancement [36] and strongly improves the sensitivity for acute vascular alternative pathologies [50]. On the other hand, extralumination of rectal contrast is detectable at a perforation site only in a minority of cases, according to our previous studies [35,37]. We assume that the obscuration of mucosal enhancement by intraluminal contrast outweighs the advantages of improved colonic distension, and rectal contrast ultimately does not facilitate the appraisal of colonic inflammation [34].

This study has some limitations. First, the retrospective design carries a risk of selection bias, which we tried to mitigate by including consecutive cases. Second, in most cases, we used the original radiology reports (based on the full set of images) from different radiology consultants as the gold standard to keep our study close to everyday clinical practice, but this may have caused some variability in the quality of our gold standard. Third, the primary categories of the assessment did not strictly follow any established classification system. As suggested by some authors, the original reports that served as the gold standard of our study used a morphological description rather than a classification of disease stage [51,52]. Only a limited number of several known CT signs of diverticulitis [53] were systematically assessed. Fourth, the size of the study population in our and similar monocentric studies provides a relatively small number of rare manifestations and complications of the disease. A detailed assessment of these could be achieved with multicentric studies. Fifth, the investigation had to be limited to a comparison of the non-contrast examinations to CTs with both intravenous and intraluminal contrast since our standard in-house protocol consisted of these two acquisitions. Future studies with different CT protocols could allow for separate evaluations of the benefits of intravenous and intraluminal contrast. This could be helpful for further optimization of CT examination protocols for suspected colonic diverticulitis.

## 5. Conclusions

The present study shows that the CT protocol for the diagnosis of sigmoid diverticulitis can be simplified by omitting intravenous and rectal contrast without significantly reducing diagnostic accuracy. An additional CT series after administration of intravenous contrast and/or a rectal contrast enema could be reserved for cases that are still equivocal after non-contrast imaging or when certain alternative diagnoses have to be ruled out. Reduced use of contrast agents has potential implications on patient comfort and safety, CT examination time, radiation dose, and expenditures for contrast agents.

## Figures and Tables

**Figure 1 diagnostics-15-00029-f001:**
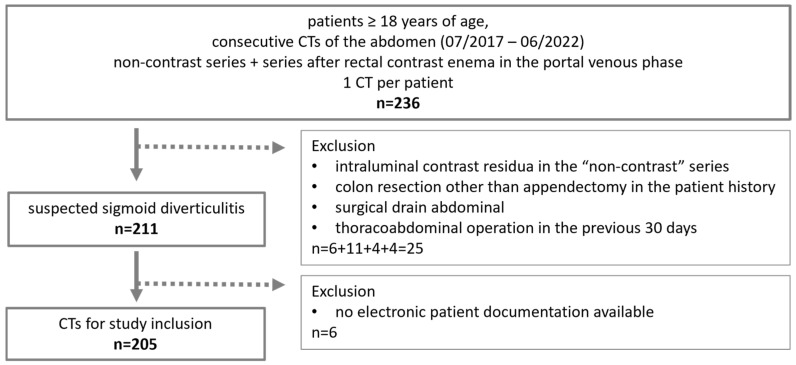
Flowchart of the patient population.

**Figure 2 diagnostics-15-00029-f002:**
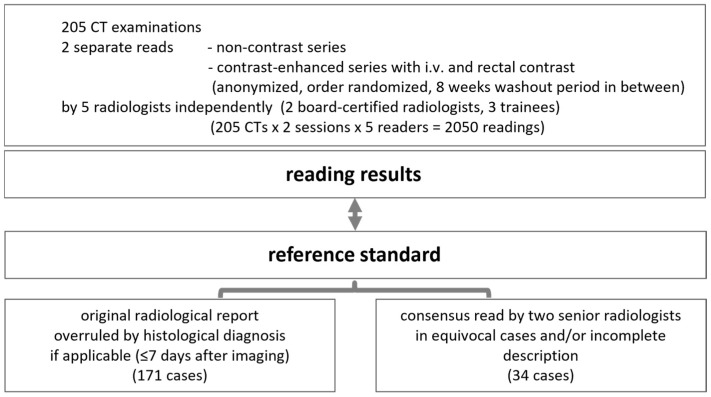
Study design for readings and reference standards.

**Figure 3 diagnostics-15-00029-f003:**
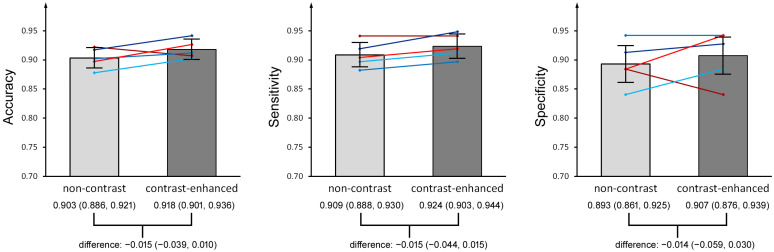
Bar chart showing the reader performance for diagnosing colonic diverticulitis in non-contrast and contrast-enhanced CT. The bars indicate accuracy, sensitivity, and specificity values for all readers (whiskers: 95% confidence intervals, CI). The lines indicate individual reader values: dark red and red (R1 and R2): board-certified radiologists; medium, dark, and light blue (R3, R4, R5): residents. Below the bars are the differences in accuracy, sensitivity, and specificity (with 95% CI). Note that the CIs of all differences shown include the 0.

**Figure 4 diagnostics-15-00029-f004:**
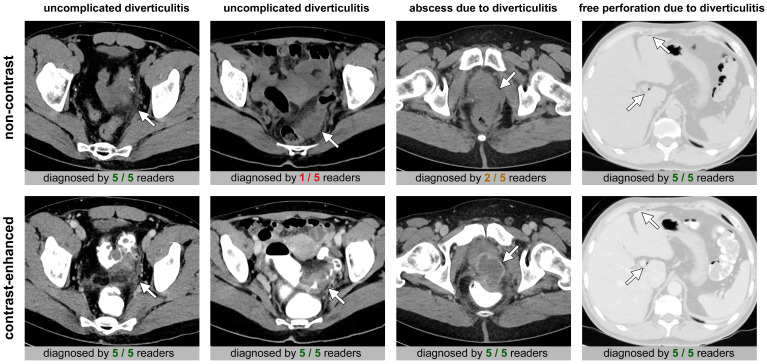
Examples of diverticulitis cases without and with complications. Upper row: representative non-contrast CTs; bottom row: corresponding contrast-enhanced CT images. The number of accurate readings is given below each CT image. Left: In most cases, colonic diverticulitis was correctly diagnosed both in non-contrast and contrast-enhanced CT (arrow: thickened diverticulum with surrounding fat stranding). Middle left: Difficult case where diverticulitis was missed in non-contrast CT by the majority of readers (arrow: thickened segment of sigmoid colon with pericolic fat stranding). Middle right: The sensitivity for abscesses was slightly lower for non-contrast compared to contrast-enhanced CT (arrow: deep pelvic abscess). Right: Free peritoneal perforations were usually correctly diagnosed both in non-contrast and contrast-enhanced CT (arrow: free perihepatic air bubbles).

**Table 1 diagnostics-15-00029-t001:** Prevalences of colonic diverticulitis, contained and free peritoneal perforations, and abscesses.

		Reference Standard	All Readers	R1	R2	R3	R4	R5
		%	%	%	%	%	%	%
diverticulitis	n.c.	66.3	63.9	66.3	63.9	60.5	63.9	64.9
	c.e.		64.4	67.8	62.9	61.5	65.4	64.4
perforation	n.c.	28.3	21.9	19.0	21.5	20.0	21.5	27.3
	c.e.		26.4	22.0	25.4	28.3	29.8	26.8
contained perforation	n.c.	23.4	17.0	15.1	16.6	15.1	15.6	22.4
	c.e.		20.9	17.5	20.0	22.9	22.0	22.0
free perforation	n.c.	4.9	4.9	3.9	4.9	4.9	5.9	4.9
	c.e.		5.6	4.4	5.4	5.4	7.8	4.9
abscess	n.c.	18.1	13.9	10.7	14.6	15.6	9.8	18.5
	c.e.		17.6	15.1	16.1	25.4	14.2	17.1

Prevalences in non-contrast (n.c.) CT and contrast-enhanced (c.e.) CT for the reference standard, for all readers pooled and for individual readers (R1–R5).

**Table 2 diagnostics-15-00029-t002:** Reader performance in diagnosing colonic diverticulitis, perforation, and abscess in non-contrast and contrast-enhanced CT.

Diverticulitis		R1		R2		R3		R4		R5	
Per Reader			95% CI		95% CI		95% CI		95% CI		95% CI
accuracy	n.c.	0.922	(0.876, 0.955)	0.898	(0.848, 0.936)	0.902	(0.853, 0.939)	0.917	(0.871, 0.951)	0.878	(0.825, 0.920)
	c.e.	0.907	(0.859, 0.943)	0.927	(0.882, 0.959)	0.912	(0.865, 0.947)	0.942	(0.900, 0.969)	0.902	(0.853, 0.939)
sensitivity	n.c.	0.941	(0.887, 0.974)	0.904	(0.842, 0.948)	0.882	(0.816, 0.931)	0.919	(0.860, 0.959)	0.897	(0.833, 0.943)
	c.e.	0.941	(0.887, 0.974)	0.919	(0.860, 0.959)	0.897	(0.833, 0.943)	0.949	(0.897, 0.979)	0.912	(0.851, 0.954)
specificity	n.c.	0.884	(0.784, 0.949)	0.884	(0.784, 0.949)	0.942	(0.858, 0.984)	0.913	(0.820, 0.967)	0.841	(0.733, 0.918)
	c.e.	0.841	(0.733, 0.918)	0.942	(0.852, 0.980)	0.942	(0.858, 0.984)	0.928	(0.839, 0.976)	0.884	(0.784, 0.949)
all readers		diverticulitis		perforation: pooled		perforation: contained		perforation: free		abscess	
			n.c. − c.e.		n.c. − c.e.		n.c. − c.e.		n.c. − c.e.		n.c. − c.e.
accuracy	n.c.	0.903	−0.015	0.865	−0.024	0.848	−0.025	0.982	−0.001	0.872	−0.023
	c.e.	0.918	(−0.039, 0.010)	0.890	(−0.053, 0.004)	0.873	(−0.055, 0.005)	0.981	(−0.011, 0.013)	0.896	(−0.051, 0.004)
sensitivity	n.c.	0.909	−0.015	0.575	−0.150	0.538	−0.138	0.820	−0.060	0.530	−0.168
	c.e.	0.924	(−0.044, 0.015)	0.725	(−0.198, −0.051) †	0.675	(−0.224, −0.051) †	0.880	(−0.203, 0.083)	0.697	(−0.266, −0.069) †
specificity	n.c.	0.893	−0.014	0.951	0.015	0.943	−0.009	0.991	−0.004	0.948	0.008
	c.e.	0.907	(−0.059, 0.030)	0.936	(−0.009, 0.039)	0.934	(−0.015, 0.033)	0.987	(−0.005, 0.013)	0.939	(−0.014, 0.030)

Accuracies, sensitivities, and specificities with their 95% confidence intervals (CIs) or their differences in non-contrast (n.c.) and contrast-enhanced (c.e.) CT for individual readers (R1–R5) for the diagnosis of colonic diverticulitis (present/absent) and for the diagnosis of colonic diverticulitis, perforation (all perforations pooled/separate analysis for contained and free perforations), and abscess (present, absent) for all readers pooled; 95% CI of the differences, not including the 0, are marked with †.

**Table 3 diagnostics-15-00029-t003:** Interrater agreement.

All Readers		Krippendorff’s α	CI 95%	pa	pe
diverticulitis	n.c.	0.661	(0.616, 0.707)	0.844	0.539
	c.e.	0.651	(0.605, 0.696)	0.840	0.541
perforation	n.c.	0.484	(0.431, 0.538)	0.810	0.632
	c.e.	0.542	(0.491, 0.594)	0.815	0.597
abscess	n.c.	0.384	(0.313, 0.455)	0.844	0.747
	c.e.	0.484	(0.415, 0.552)	0.857	0.723

Krippendorff’s α with their 95% confidence intervals (CI), overall percent agreement (pa), and percent chance agreement (i.e., percent agreement expected by chance; pe) for the diagnosis of diverticulitis, perforation, and abscess in non-contrast (n.c.) and contrast-enhanced (c.e.) CTs for all readers. (0.00–0.20 = slight, 0.21–0.40 = fair, 0.41–0.60 = moderate, 0.61–0.80 = substantial, and 0.81–1.00 = almost perfect agreement [29]).

**Table 4 diagnostics-15-00029-t004:** Prevalences of alternative diagnoses in the reference standard and percentages of reader ratings with their successful identification.

	Prevalence of Alternative Diagnoses		Identification of the Alternative Diagnosis			
	In Reference Standard		In % of Readings per Diagnosis (*n* × 5)			
	*n*	% of All Cases	In n.c.	In c.e.	Only in n.c.	Only in c.e.
non-diverticulitis-associated colitis/ileitis	9	4	71	82	7	18
appendicitis	5	2	92	96	4	8
ileus	4	2	70	70	10	10
perforation	5	2	92	100	0	8
tumor/mesenteric lymphadenopathy	4	2	75	85	0	10
mesenteric thrombosis/embolism	3	1	7	87	0	80
abscess or tumor of the adnexes	3	1	67	80	13	27
urinary tract obstruction/infection	7	3	74	74	9	9
other	5	2	56	60	12	16
sum:	45	22				

Left: prevalences of alternative diagnoses for abdominal pain and/or elevated inflammatory markers in the reference standard. Right: percentages of readings with successful identification of each alternative diagnosis: identified in non-contrast (n.c.) CT, in contrast-enhanced (c.e.) CT, only in non-contrast CT (missed in contrast-enhanced CT), only in contrast-enhanced CT (missed in non-contrast CT).

## Data Availability

The data presented in this study are available on request from the corresponding author due to ethical reasons.

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
