# Peer review of "Comparison of Non-Contrast CT vs. Contrast-Enhanced CT with Both Intravenous and Rectal Contrast Application for Diagnosis of Acute Colonic Diverticulitis: A Multireader, Retrospective Single-Center Study"

_diagnostics, 2024, doi:10.3390/diagnostics15010029_

Round 1

Reviewer 1 Report (Previous Reviewer 2)

Comments and Suggestions for Authors

Dear authors,

although you resubmitted your article, my review is of the same content. Since all requested requests were explained and corrected, I believe as a reviewer that the quality of the work has been significantly improved, ambiguities have been improved. In my opinion, the work is much better now and it is possible to accept it.

Best Regards

My previus review  (reviewer 2). 

Dear authors, congratulations on the article.

In order to improve the quality of work, I recommend the following corrections.

Line 28.higher sensitivities. Please write is this significant or non significant deference?

Line 61. Please in the section material and methodes write the type of CT (256-MSCT or another), and manufacturer of CT devices.

Line 63. please write the type of contrast agent and manufacturer

I reccomend citation of this articles: 

Tiralongo, F.; Di Pietro, S.; Milazzo, D.; Galioto, S.; Castiglione, D.G.; Ini’, C.; Foti, P.V.; Mosconi, C.; Giurazza, F.; Venturini, M.; et al. Acute Colonic Diverticulitis: CT Findings, Classifications, and a Proposal of a Structured Reporting Template. Diagnostics 2023, 13, 3628. https://doi.org/10.3390/diagnostics13243628

Please emphasize in the introduction which diagnostic method of first choice and what the European guidelines recommend, and please add new references regarding this.

Do you consider the number of CT findings to be a limitation of the study?

I advise you to change the title, my proposal: Comparison of non-contrast CT vs contrast-enhanced CT with both intravenous and rectal contrast application for diagnosis of acute colonic diverticulitis:A multireader, retrospective single-center study

Were all CT scans done on the same device with the same number of slices?

Have you experienced any complications related to your constart allergy or constart nephropathy? Write it down.

Also state how much time it takes to perform CT without contrast and with contrast per patient.

Best Regards

Reviewer 2 Report (Previous Reviewer 1)

Comments and Suggestions for Authors

The authors have revised the manuscript according to the reviewers' comments, and the revised manuscript is satisfactory.

This manuscript is a resubmission of an earlier submission. The following is a list of the peer review reports and author responses from that submission.

Round 1

Reviewer 1 Report

Comments and Suggestions for Authors

The authors investigated the non-inferiority of non-contrast CT by comparing to contrast-enhanced CT with both intravenous and rectal contrast application for diagnosis of acute colonic diverticulitis. This article is well written, and interesting and has useful information.

I have some comments for this manuscript.

# Introduction

OK.

# Materials and Methods.

OK.

# Results

OK.

# Figure

The authors wrote the figure legends for Figure 4 as follows; “Figure 4. Examples of diverticulitis cases without and with complications. Upper row: representative n.c. CTs, bottom row: corresponding c.e. CT images. The number of accurate readings is given below each CT image. Left: In most cases, colonic diverticulitis was correctly diagnosed both in n.c. and c.e. CT. Middle left: Difficult case where diverticulitis was missed in n.c. CT by the majority of readers. Middle right: The sensitivity for abscesses was slightly lower for n.c. compared to c.e. CT. Right: Free peritoneal perforations were usually correctly diagnosed both in n.c. and c.e. CT.”. 

According to Figure 4, the middle left case was accurately diagnosed by 5/5 readers on n.c. CT, and 1/5 readers on c.e. CT. Therefore, the explanation of “difficult case where diverticulitis was missed in n.c. CT by the majority of readers.” would be not correct.

Discussion:

As the authors also mention in the limitations section, this study was limited to a comparison of non-contrast examinations with CTs that included both intravenous and intraluminal contrast. If possible, it would be beneficial to provide further discussion on this aspect.

Reviewer 2 Report

Comments and Suggestions for Authors

Dear authors, congratulations on the article.

In order to improve the quality of work, I recommend the following corrections.

Line 28.higher sensitivities. Please write is this significant or non significant deference?

Line 61. Please in the section material and methodes write the type of CT (256-MSCT or another), and manufacturer of CT devices.

Line 63. please write the type of contrast agent and manufacturer

I reccomend citation of this articles: 

Tiralongo, F.; Di Pietro, S.; Milazzo, D.; Galioto, S.; Castiglione, D.G.; Ini’, C.; Foti, P.V.; Mosconi, C.; Giurazza, F.; Venturini, M.; et al. Acute Colonic Diverticulitis: CT Findings, Classifications, and a Proposal of a Structured Reporting Template. Diagnostics 2023, 13, 3628. https://doi.org/10.3390/diagnostics13243628

Please emphasize in the introduction which diagnostic method of first choice and what the European guidelines recommend, and please add new references regarding this.

Do you consider the number of CT findings to be a limitation of the study?

I advise you to change the title, my proposal: Comparison of non-contrast CT vs contrast-enhanced CT with both intravenous and rectal contrast application for diagnosis of acute colonic diverticulitis:A multireader, retrospective single-center study

Were all CT scans done on the same device with the same number of slices?

Have you experienced any complications related to your constart allergy or constart nephropathy? Write it down.

Also state how much time it takes to perform CT without contrast and with contrast per patient.

Best Regards
